# Effects of sea ice and wind speed on phytoplankton spring bloom in central and southern Baltic Sea

**Ove Pärn** [1]*, **Gennadi Lessin**[2], **Adolf Stips**[1]

**1** European Commission, Joint Research Centre (JRC), Ispra, Italy, **2** Plymouth Marine Laboratory, Plymouth, United Kingdom

* Ove.PARN@ext.ec.europa.eu

## Abstract

In this study, the effects of sea ice and wind speed on the timing and composition of phytoplankton spring bloom in the central and southern Baltic Sea are investigated by a hydrodynamic–biogeochemical model and observational data. The modelling experiment compared the results of a reference run in the presence of sea ice with those of a run in the absence of sea ice, which confirmed that ecological conditions differed significantly for both the scenarios. It has been found that diatoms dominate the phytoplankton biomass in the absence of sea ice, whereas dinoflagellates dominate the biomass in the presence of thin sea ice. The study concludes that under moderate ice conditions (representing the last few decades), dinoflagellates dominate the spring bloom phytoplankton biomass in the Baltic Sea, whereas diatoms will be dominant in the future as a result of climate change i.e. in the absence of sea ice.

## Introduction

The Baltic Sea has suffered historically from anthropogenic river borne nutrient loads due to intensified agriculture and waste-water discharges since the 1950s [1] and the phytoplankton biomass in Kieli Bay tripled in the 1960s [2]. Long nutrient residence times and high buffer capacity of the system have resulted in slow responses to nutrient load reductions. For example, the pan-European marine modelling ensemble study evaluated the impact of freshwater nutrient control measures on marine eutrophication indicators. When anthropogenic riverine nutrients were reduced by 10%, the chlorophyll-a concentration decreased by 1% on an average [3]. Concentrations of nutrients have accumulated to such a high level that human effects on nutrient reduction have been barely noticeable in recent decades.

A unique combination of physical, chemical, and biological factors shape the marine ecosystem. Due to the complexity of the system, it is difficult to distinguish between the impact of climate change and human eutrophication on the ecosystem. Studies that clarify the role of humans or climate change in marine life are essential to achieving a healthy marine ecosystem [4]. The Baltic Sea, a seasonally ice-covered semi-enclosed sea, is sensitive to climate change and is affected by various human-induced environmental pressures caused by shipping,

**Data Availability Statement:** All relevant data are within the paper and its Supporting information files.

**Funding:** The authors received no specific funding for this work.

**Competing interests:** The authors have declared that no competing interests exist.

eutrophication, as well as chemical and plastic pollution [5]. This paper analyses the causes for the dramatic decrease of the diatom standing stock and a change towards the dominance of autotrophic dinoflagellates in the Baltic since the late 1980s [6–9].

Physical factors like light and stratification are key influences controlling phytoplankton spring bloom [10]. The seasonal ice cover plays a crucial role in setting time frames for the primary production, thus affecting the seasonality of ecological processes. Ice cover is one of the major indicators in determining the severity of winter in the Baltic Sea. It also affects underwater light conditions by limiting the amount of light transmitted through the surface and modifying water mixing and nutrient circulation under the ice [11].

The Baltic Sea ice season lasts 5–7 months between November to May. During a mild winter, ice occurs only in the Bothnian Bay and the easternmost Gulf of Finland, while the whole Baltic Sea is covered with ice during a cold winter [12]. The maximum extent of the ice cover in the Baltic Sea is normally reached towards late February or early March. In the southern part of the Baltic sea, ice conditions vary extensively from one year to another. The maximum ice extent in the Baltic Sea has been steadily decreasing for the last two centuries [12]. Global warming, induced by the climate change, has reduced the thickness of ice and shortened the ice season [13].

Studies on phytoplankton spring bloom have been increasingly gaining importance in the context of global climate change. It has been reported that the changing environmental conditions have reduced the coverage and thickness of sea ice and increased the water temperatures in the Baltic Sea [14–17]. Increased precipitation in the northern Baltic catchment affects freshwater inflow and nutrient run-off [18]. The continually rising air temperature and freshwater inflow are potential factors affecting the stratification of water layers, which in turn affect the vertical transportation of oxygen and planktonic life forms particularly in spring [19]. Physical conditions provide a foundation for biological activity. During winters there is low nutrient consumption under the ice and the water body will be enriched with nutrients. The phytoplankton biomass in the Baltic Sea is low during winters; it increases from March and grows continuously for up to 2 months [20–22]. Studies have reported that the warming of the Baltic Sea has caused temporal shifts in the distribution of phytoplankton during the highly productive spring season. An early onset and increased duration of spring bloom have been observed due to warming of the sea environment [23–26]. In addition, long-term monitoring data suggests that there may be a shift in the composition of the phytoplankton biomass from a diatom to dinoflagellate-dominated assemblages during spring in some areas of the Baltic Sea [2, 27] because of ongoing climate change [20]. Stratification of the water column is generally a prerequisite for most dinoflagellate blooms to develop in temperate areas [27, 28]. Wind is one of the factors governing vertical mixing within the euphotic layer (average 9.6 m in Baltic [29]) and in stagnant water, the diatoms being non-motile and heavier particles start sinking quickly whereas the dinoflagellates are motile and can therefore position themselves vertically. Dinoflagellate blooms are observed in coastal waters under calm wind conditions enhancing the water column stratification [30]. Spilling et al. [9] analysed the shift from diatom dominance to dinoflagellate dominance in parts of the Baltic Sea during spring bloom and its potential effect on biogeochemical cycling. According to their research, the shift towards dinoflagellate dominance increased the sinking of organic matter which alleviated the issue of eutrophication and improved the environmental status of the Baltic Sea.

Diatoms constitute the major component of phytoplankton spring bloom in most coastal ecosystems, whereas vernal phytoplankton communities in the Baltic Sea are characterised by the co-occurrence and often dominance of cold-water dinoflagellates [27, 31]. Factors promoting the success of dinoflagellates in dominating the spring phytoplankton community are insufficiently understood because dinoflagellates are considered to be inferior to diatoms as

competitors because of their relatively low growth rates and nutrient uptake capacities [32]. Although dinoflagellate-dominated spring bloom is uncommon elsewhere, it regularly occurs in the Baltic Sea and its frequency has increased over the years [9]. This paper examines the effects of sea ice and wind speed on phytoplankton spring bloom and describes the potential mechanisms influencing the blooming of dinoflagellates or diatoms. We hypothesize that thinner sea ice or low wind speed (low turbulence in the euphotic zone) ensures early seeding and accumulation of dinoflagellates, which allows them to outcompete the faster growing diatoms. To test the validity of this mechanism we simulate ice-free conditions, where the determining factor was wind-induced turbulence in the euphotic layer. The effect of the mechanism on diatoms and dinoflagellates was tested with observational data.

## Materials and methods

A coupled hydrodynamic–biogeochemical model was used to estimate the effects of sea ice and wind speed on phytoplankton composition during spring between 1 March and 10 May in the relatively severest winters of 2010, 2011 and 2013. Recent years have mainly observed mild winters in the Baltic sea. The results of the model in the presence and the absence of sea ice were compared. All the runs used the same initial and boundary conditions; however, sea ice conditions were varied.

### Model description and set-up

The model simulations were performed by coupled three-dimensional hydrodynamic model GETM (General Estuarine Transport Model; https://getm.eu/) [33, 34] and biogeochemical model ERGOM (Ecological Regional Ocean Model; www.ergom.net) implemented within the Framework for Aquatic Biogeochemical Models [35]. The ERGOM model version applied in this study included three functional groups of phytoplankton—diatoms, dinoflagellates, and cyanobacteria; bulk zooplankton group; nitrate, ammonium, and phosphate nutrients; dissolved oxygen; pelagic and benthic detritus; and iron-bound phosphorus in sediments and water. The corresponding set-up was previously developed and used by Lessin et al. [36, 37].

The model domain covered the entire Baltic Sea area, with an open boundary in northern Kattegat. Bathymetry was interpolated to a model grid with a horizontal resolution of $2 \times 2$ nm. Twenty-five σ layers were applied in the vertical [36]. The period modelled was between 1 September and 10 July in 2010, 2011 and 2013. During the first 6 months of simulation, only hydrodynamics was modelled to ensure that the initial biochemical conditions before spring bloom were identical for all the model runs, highlighting how ice affects plankton development in the spring season. The default dinoflagellate maximum growth rate in the model was 0.4 $d^{-1}$, which led to very low small-cell phytoplankton concentrations. After extensive calibration for the study region, the growth rate was adjusted to 0.7 $d^{-1}$, the same value was used by the Leibniz Institute for Baltic Sea Research, Warnemünde in its ecosystem model [38]. The default diatom maximum growth rate was set at 1.3 $d^{-1}$, and the sinking velocity of diatoms was 0.5 m/d. Minimum irradiances for dinoflagellates and diatoms were 50 $W/m^2$ and 35 $W/m^2$, respectively.

The initial distributions of water temperature and salinity for September were interpolated to the model grid from monthly climatological data [39]. The prescribed salinity and temperature distributions at the open boundary were interpolated using monthly climatological data. Typical concentrations of biogeochemical variables were prescribed uniformly within the model domain.

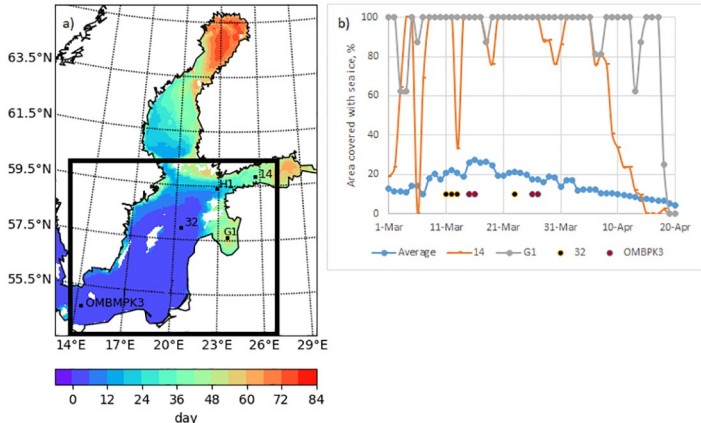

**Fig 1.** (a) Map of the Baltic Sea indicating stations (located in different physical-biogeochemical conditions) where modelled water densities and phytoplankton concentrations were analysed. The colour scale indicates the number of ice days since 1 March 2013. The black border represents the sea area for which the average is obtained; (b) The concentration of sea ice at stations and the average coverage over the area.

Six-hourly meteorological forcing data (ERA-Interim) obtained from the European Centre for Medium-Range Weather Forecasts (http://www.ecmwf.int) was applied. The model considered land-based run-off and nutrient loads incorporated into 20 major rivers [36].

The following scenarios were modelled to estimate the effect of sea ice on phytoplankton spring bloom:

1. Simulation with sea ice (ice run, as reference scenario).

2. Simulation without sea ice (no ice run).
   And to test the validity of the proposed mechanism

3. Simulation with sea ice and 10% of wind speed compared to the reference speed recorded during five days from 06–10 April, 2013 (wind reduction scenario). From the results obtained, we calculated the increase rate (%) of dinoflagellates and diatoms due to the wind reduction: [(reduction scenario—reference scenario)/reference scenario*100].

Diatom and dinoflagellate concentrations and water density at four stations (Fig 1) were analysed according to the scenarios in the central and southern Baltic Sea between 1 March 2013 and 10 May 2013. Stations were selected according to gradient in physical-biogeochemical conditions and specific differences in ice cover (Fig 1a), the boxes had an area of 6 × 6 nm.

## Model validation

The main validation of the ocean and biochemisry component of the coupled model is provided in the report on the biogeochemical model setup for the Baltic Sea and its applications [40]. Hydrodynamic features such as salinity, temperature, and surface elevation were reproduced well. From a comparison of the modelled SST with satellite SST data the averaged bias over space and time was calculated to be 0.7˚C. Stations of observations (BED, http://nest.su.se), root-mean-square error of sea surface and bottom salinity were between 0.3–1.7 PSU. The root-mean-square error of sea level elevation at Landsort measured data from Permanent Service for Mean Sea Level was 198 mm in 2005–2012. The model was able to reproduce the ice-covered areas as well as the interannual variability. All modelled eutrophication indicators—chlorophyll-a, oxygen, nitrate, and phosphate—followed the dominant seasonal cycles. The

model simulated chl-a data more accurately in the southern Baltic Sea (RMSE = 0.9), and root-mean-square error was larger in the Gulf of Finland, 4 mg m$^{-3}$. However, the model underestimated the phytoplankton biomass (chlorophyll-a concentration) in most parts of the Baltic Sea and overestimated the oxygen content compared to the observations which might be a consequence of the too low phytoplankton biomass.

The nitrate concentration was overestimated by the model when compared to BED data (http://nest.su.se), except North Baltic, but underestimated when compared to HELCOM data. Phosphate was overestimated in the Gulf of Bothnia and the Gulf of Finland; however, it was underestimated in the rest of the Baltic Sea. The excess phosphate in the Gulf of Finland is most probably causing the high Chlorophyll-a concentration here. The maximum values of chlorophyll-a in April 2013 are shown in S1 Table, stations shown in Fig 1.

## Ice and phytoplankton data

Data on the daily ice concentrations over the Baltic Sea was provided by the Copernicus Marine Environment Monitoring Service (http://marine.copernicus.eu/). The ice concentration data originated from the Swedish Ice Service (http://www.smhi.se/oceanografi/istjanst/havsis_en.php), and the radiometric satellite observations obtained using the sensors AVHRR, ATSR-1, ATSR-2, and AATSR were used in this study. These data were used to calculate the ice cover characteristics of the Baltic Sea for the ice season by interpolating the ice concentration data to the model grid. Areas with an ice concentration of <40% were considered ice-free (open water). Ice was considered to be present in the model grid cells if the ice concentration was ≥40%.

In the presence of ice:

- the surface temperature of the water is equal to the freezing point

- the wind stress that causes circulation is 0

- light conditions change. In the case of sea ice, $PAR_i = 0.7^*PAR$, where PAR is photosynthetically active radiation and $PAR_i$ is PAR under ice [41].

Swedish National Monitoring data includes various biological and physio-chemical parameters collected at designated stations. For this study, data on diatoms and dinoflagellates biomass was downloaded from the national marine environmental monitoring database (SHARKweb).

The IOW data are available on request from the Oceanographic Database of IOW (IOWDB) (https://www.io-warnemuende.de/en_iowdb.html).

Lithuanian marine monitoring data as well as diatoms and flagellates biomass values were selected and systematized by Irina Olenina.

## The Dia/Dino index

The European Parliament has adopted the Marine Strategy Framework Directive (MSFD) to protect the marine environment across Europe, which is based on a set of environmental descriptors. One of the supporting descriptors is the Dia/Dino index, which is a pre-core indicator, as decided by the Baltic Marine Environment Protection Commission (Helsinki Commission [HELCOM]; http://www.helcom.fi). HELCOM uses indicators with quantitative threshold values to evaluate the progress towards the goal of achieving an acceptable environmental status of the Baltic Sea. The index is based on seasonal diatom and dinoflagellate biomass values and reflects dominance patterns in the biomass of phytoplankton spring bloom. It describes the relative percentage of diatoms within the total biomass and provides a proxy to

estimate the entry of nutrients into the pelagic or benthic food web: if diatoms are dominant, their rapid sinking reduces the availability of food stock for zooplankton, but delivers abundant food to zoobenthos [5]. Changes in diatom or dinoflagellate dominance affect the pelagic and benthic food web dynamics due to differences in their quality as food sources and the timings of their occurrence [2]. A low Dia/Dino index indicate silicate limitation caused by eutrophication [5].

The Dia/Dino index [2] presents the relative percentage of diatoms and was calculated as follows:

$$\frac{Dia}{Dino} index = \frac{Concentration of Diatoms \left[\frac{mgChl}{m^3}\right]}{Concentration of Diatoms + Concentration of flagellates \left[\frac{mgChl}{m^3}\right]} \tag{1}$$

It was used to calculate the spring averages (10 March–05 May) of the phytoplankton concentration in Chl units for the surface layer (0-10m).

## Results

### Effect of sea ice on the composition of spring bloom in the central Baltic Sea

The average modelled concentrations (upper 10 m) of dinoflagellates and diatoms over the central Baltic Sea ice covered area (Fig 1, the area inside the black rectangle, if ice days > 2) during spring 2013 are presented in Fig 2. Diatom bloom reached its maximum concentrations in simulations without sea ice. Variations in the dinoflagellate concentration were opposite to those observed in the diatom concentration; the presence of sea ice favoured higher dinoflagellate concentration during ice-free conditions. The peak bloom concentration of dinoflagellates decreased by more than two times, during simulations in the absence of sea ice. The ice-free area results revealed that, on average, the diatom concentration was higher (up to 5mg Chl/m$^3$), whereas the dinoflagellate concentration did not exceed 0.45mg Chl/m$^3$ in the ice-free area.

The maximum diatom concentration was 3.5 mg Chl/m$^3$ in simulation with ice and 5.2 mg Chl/m$^3$ in simulation without ice (Fig 2). Corresponding concentrations of dinoflagellates

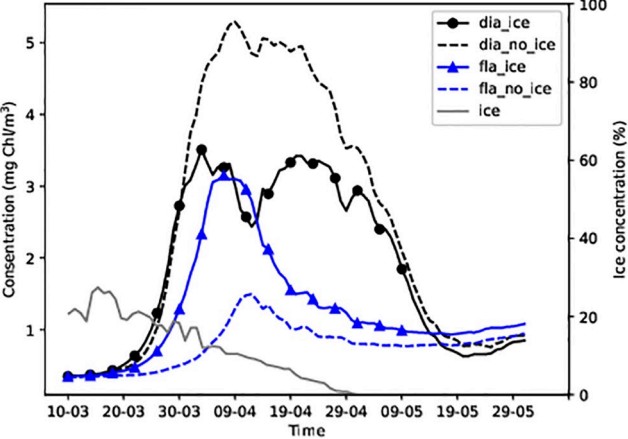

**Fig 2. Average diatom and dinoflagellate concentrations over the central Baltic Sea in spring 2013 in the biggest area (Fig 1) over which sea ice was present during the season.** Continuous lines represent the results of simulations with sea ice whereas broken lines represent the results of simulations without sea ice.

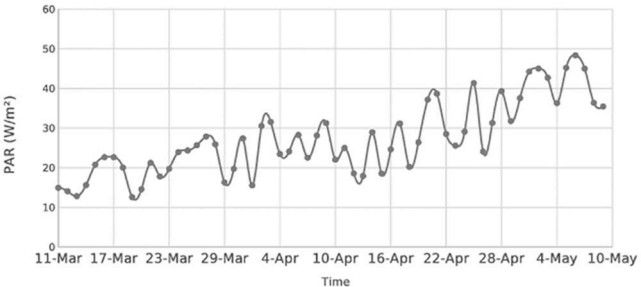

**Fig 3. Daily average of photosynthetically active radiation (PARi) on the water surface under ice.**

were 3.2 and 1.4 mg Chl/m³, respectively. The phytoplankton started growing on 24–25 March (Fig 2) when the daily average $PAR_i$ exceeded 24 W/m³ under sea ice (Fig 3).

In the model, the growth of dinoflagellates, but not diatoms, directly depends on the temperature of water [42]. We examined the effects of differences in surface temperatures in the two simulations on the dinoflagellate concentration. Although the difference between average water temperatures was—0.8°C, the resulting temperature-dependent difference in the growth rate of flagellates was only up to 3%. Thus, the temperature difference was not sufficiently large to cause the difference in the phytoplankton growth rates.

## Dinoflagellates and diatoms at stations

Stations 14 in the Gulf of Finland and G1 in the Gulf of Riga were covered with sea ice in the 2013 spring season (Fig 1b). Dinoflagellate-dominated spring bloom appeared only in the model simulation in the presence of sea ice at station 14 (Figs 4 and 5). By contrast, at station G1 diatom-dominated bloom appeared only in the model simulation in the absence of ice (Fig 5).

At station G1, no growth of dinoflagellates was observed in the absence of sea ice. The diatom concentration increased rapidly from 24 March when the daily average PAR reached 32 W/m² and midday air temperatures exceeded 0°C. However, a dinoflagellate-dominated bloom preceded the diatom-dominated bloom in the presence of sea ice (Fig 5). The presence

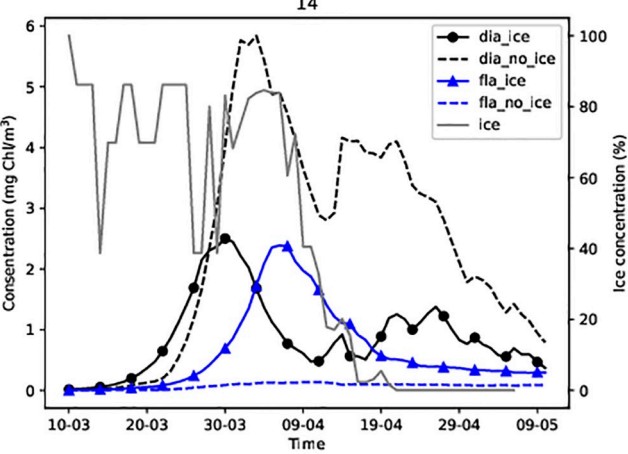

**Fig 4. Simulated diatom and dinoflagellate concentrations at station 14.**

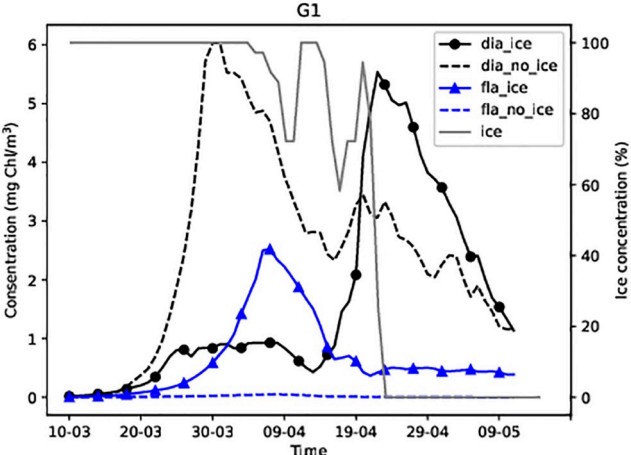

**Fig 5. Modelled diatom and dinoflagellate concentrations at station G1.**

of ice prevented the dominance of diatoms. Fig 5 shows that diatoms began to grow when ice disappeared (in mid-April), while the dinoflagellate concentration decreased.

To understand the cause of the difference in the prevalence of dinoflagellates or diatoms, we analysed the temporal dynamics of phytoplankton concentration and water density within the water column. At the shallower, ice-covered stations, the diatom concentration increased in late March–beginning of April in both simulations (Figs 6a, 6b, 7a and 7b). Compared to the water surface, diatom concentration values were higher at a depth of 10–15 m in the no ice run (Figs 6b and 7b) and (S1b–S4b Figs). However, the highest concentrations was observed at 5–10 m depth. The concentrations of diatomsin the surface layers were low in the ice run

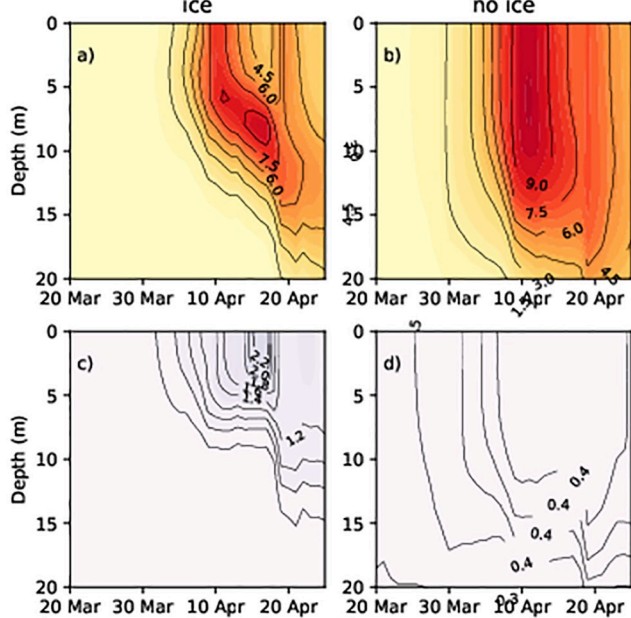

**Fig 6. Diatom and flagellate concentrations (Chl mg/m³) at station 14 in spring 2013.** (a) Diatoms (no ice run); (b) diatoms (ice run); (c) dinoflagellates (no ice run); (d) dinoflagellates (ice run).

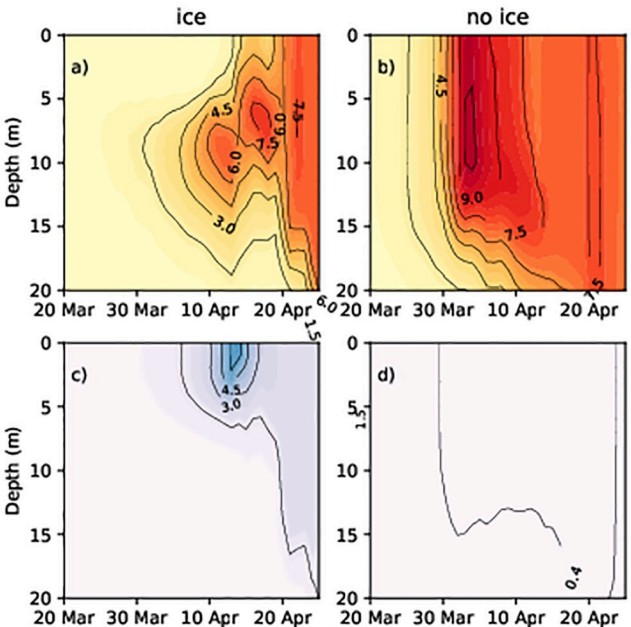

**Fig 7. Diatom and flagellate concentrations (Chl mg/m³) at station G1 in spring 2013.** (a) Diatoms (no ice run); (b) diatoms (ice run); (c) dinoflagellates (no ice run); (d) dinoflagellates (ice run).

(Figs 6a and 7a) from the beginning to mid-April while the concentrations in the surface layers were high in the ice-free run (Figs 6b and 7b).

Dinoflagellate spring bloom did not occur at stations 14 and G1 in simulations without ice (Figs 6d and 7d), while in simulations with ice, dinoflagellates developed in the upper layer (approximately 0–5 m) and diatoms lived below 6–12 m (Figs 6a, 6c, 7a and 7c). In the area partially covered with ice, the water density of surface layers is lower than that of deeper layers. Thus, lighter particles like dinoflagellates remained on the surface (and followed patterns of water mixing while) heavier diatoms sank into the deeper layers. To confirm the validity of this hypothesis, we performed a model experiment where the sinking rate of the diatoms was equivalent to the sinking rate of the dinoflagellates (both equal to 0). Therefore, the diatoms and the dinoflagellates did not sink, thus resulting in them both appearing under the ice. The maximum diatom and dinoflagellate concentration were 7 mg Chl/m³ and 1.5 mg Chl/m³ in the studied area.

Diatoms were well mixed within the upper layers during no ice run due to the turbulent mixing of the water (Figs 6b and 7b), and they thrived during spring periods as their higher growth rate enabled them to outcompete dinoflagellates. As the concentration of diatoms increases in the upper layer due to their faster growth rate, they absorb light and prevent the penetration of light to lower layers, thus, restricting the development of dinoflagellates.

In another model experiment, the light constant of limit for both dinoflagellates and diatoms were kept same. It was seen that along with dinoflagellates, diatoms also developed in the simulation without sea ice, thus confirming that light to be the limiting factor. The maximum diatom and dinoflagellate concentration were 7.5 mg Chl/m³ and 1.5 mg Chl/m³, respectively, in the studied area.

At stations 32 and OMBPK3 (Figs 8 and 9), where the presence of sea ice is a rare phenomenon (Fig 1b), the model did not show spring bloom of dinoflagellates. Diatom concentration

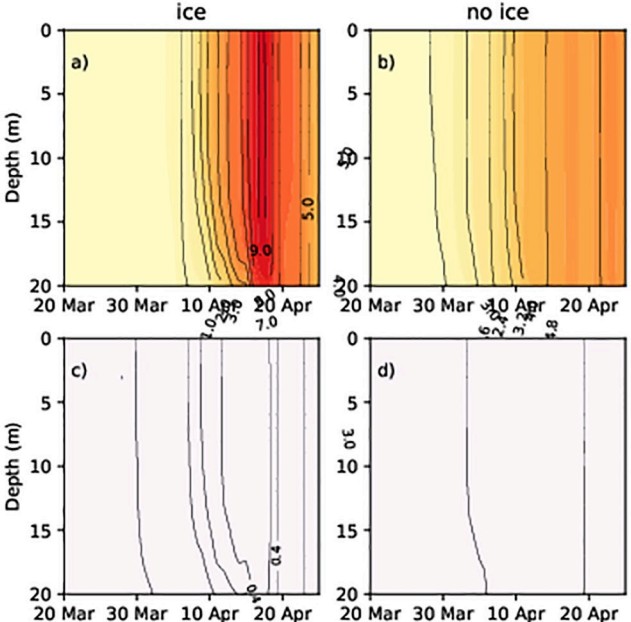

**Fig 8. Diatom and dinoflagellate concentrations (Chl mg/m$^3$) at station 32 in spring 2013.** (a) Diatoms (no ice run); (b) diatoms (ice run); (c) dinoflagellates (no ice run); (d) dinoflagellates (ice run).

patterns with ice and without ice were similar but values differed at some moments. Because ice affected the circulation of the sea as a whole, the total heat flux between the atmosphere and ocean differed and affected the distribution of nutrients as well. The changes in ice conditions in the northern part affects the state of the wider Baltic Sea.

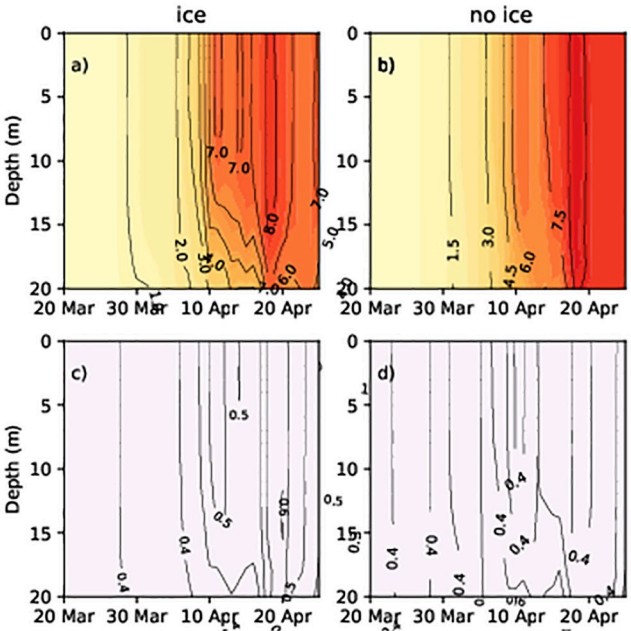

**Fig 9. Diatom and dinoflagellate concentrations (Chl mg/m$^3$) at station OMBPK3 in spring 2013.** (a) Diatoms (no ice run); (b) diatoms (ice run); (c) dinoflagellates (no ice run).

**Table 1. Dia/Dino index average between 10 March-05 May.**

| Station | Without ice | With ice |
|---|---|---|
| 14 | 0.97 | 0.5 |
| H1 | 0.98 | 0.9 |
| G1 | 0.97 | 0.61 |
| 32 | 0.99 | 0.97 |
| OMBPK3 | 0.97 | 0.97 |

## Modelled Dia/Dino index at stations

The index values calculated by using Eq (1) are listed in Table 1 below. The values of the index were high in the southern Baltic Sea, as diatoms were dominant. The Gulf of Riga and the Gulf of Finland had lower values with ice run, whereas the values of the index were high without ice; a transition to dominance of dinoflagellates over diatoms was observed in spring bloom.

Measurement data for phytoplankton biomass for spring is scarce, especially for the period after the melting of ice. Therefore, more detailed data analysis is difficult, but case studies can be performed. We have first described these kinds of cases where there are high biomasses of dinoflagellates in the observation data. The second part presents statistics between wind speed and dinoflagellate biomass where data from April and May has been used.

## Measured Dia/Dino index

**Dia/Dino index and sea ice conditions.** Sea ice information is taken from SMHI ice charts and satellite data. The data on phytoplankton biomass has been used to calculate dia/dino index of each observation and indicate presence/absence of ice cover. If the sea ice was present at or near the sampling point in the previous days, it is indicated under the column "ice cover" in S2 Table given below. Relatively high biomass values of dinoflagellates (dia/dino index < 0.5) were measured in 1991, 1996, 1998, 2001, 2003, 2004, 2005, 2009, 2010, 2011, 2013, and 2015 (S1 Table).

In S2 Table, the mean value of dia/dino index when ice occurs is 0.27, std = 0.17, while without ice it is 0.49 (std = 0.08). High relative biomass values of dinoflagellates were measured and the existence of ice was found to prevail on ten different occasions at stations A and B (S2 Table; Fig 10). All these cases had developed under ice cover, thus limiting the effect of wind on water mixing (Fig 10).

**The correlation between measured phytoplankton and wind speed in spring.** We studied the relation between wind speed and the prevalence of dinoflagellates in phytoplankton spring blooms. Phytoplankton measurements conducted between 55.3–57.3˚N and 19.8–21˚E (Fig 10, blue line) during the months of April and May from 1993–2008 by the Lithuanian EPA were used.

Out of these 16 years, dinoflagellates have had a higher biomass than diatoms during ten different years (1993, 1994, 1995, 1997, 2000, 2002, 2003, 2004, 2006, and 2008).

In the years when higher dinoflagellate biomass was measured, wind speed values were mostly lower than 5 m/s during the day preceding the measurements. Five-day average wind speed before measurement was under 5 m/s (Fig 11). The correlation between dia/dino index and the 5-day wind average before the measurement dates came out to be 0.5 (N = 16, the critical value $R_{crit}$ (P < 0.05) = 0.468). The correlation is not strong but is statistically significant. Thus, low wind speeds create physical conditions that favor faster growth of dinoflagellates. Higher wind speeds lead to greater turbulent mixing in the euphotic zone and create the conditions which are advantageous to diatoms.

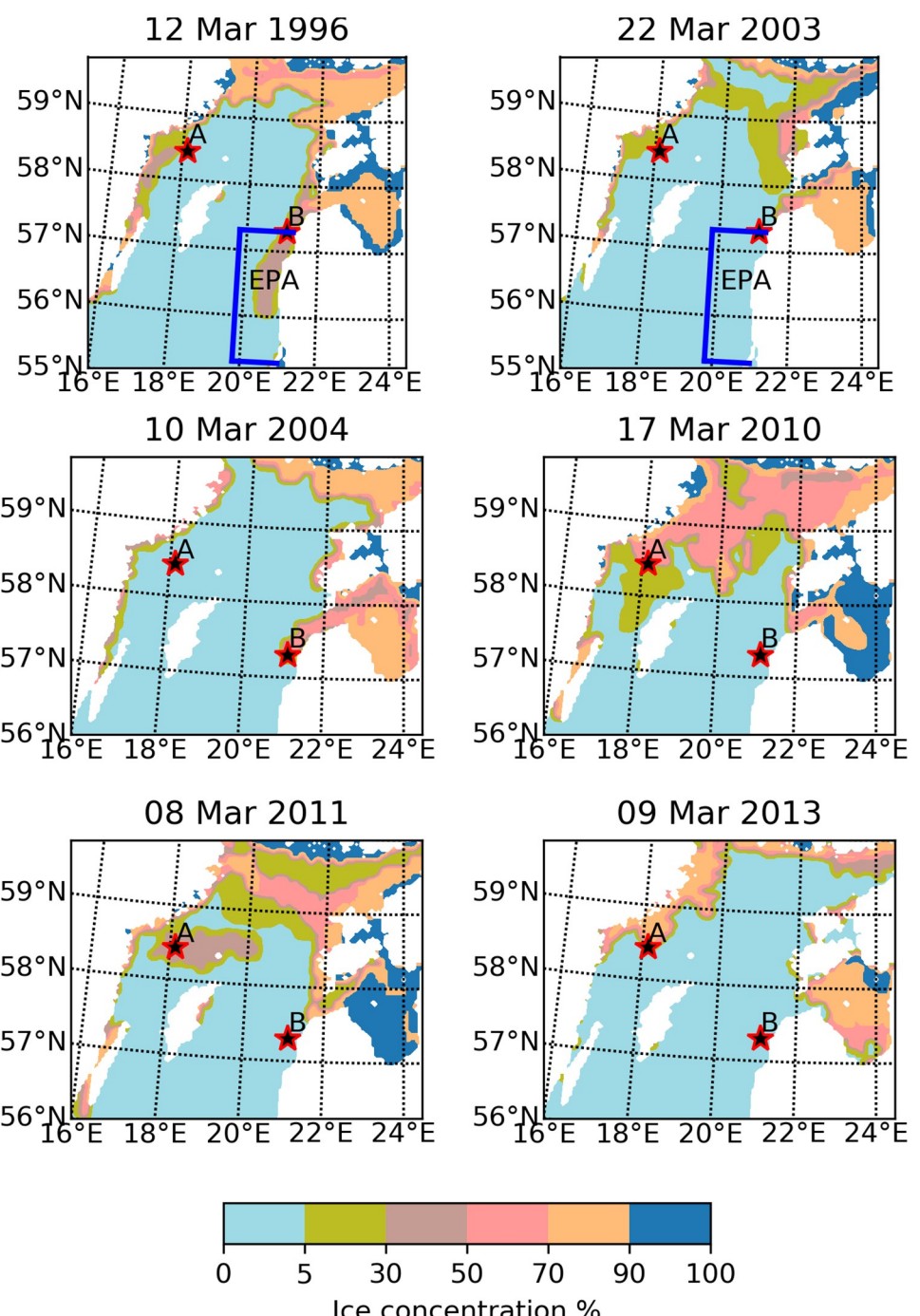

**Fig 10. Sea ice concentration on the day of phytoplankton measurement.** The measuring point is marked with a red-black star.

**The correlation between modelled phytoplankton and wind speed in spring.** Fig 12 shows the correlation between phytoplankton and a 5-day moving average of wind speed, which was calculated five days before the phytoplankton data was recorded. In the case of dinoflagellates, the correlation coefficient is negative (blue) in most areas which shows that

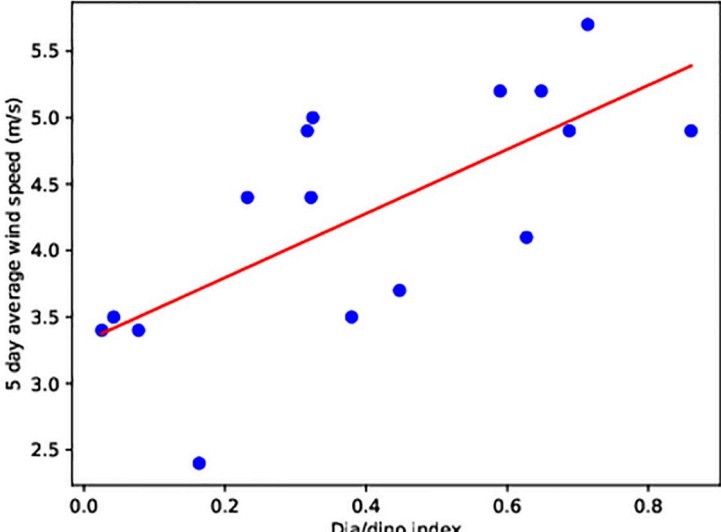

**Fig 11. Dia/dino index calculated from the measurements and five-day wind speed average from 1993–2008.** Data taken from EPA Lithuanian and ERA 5. Regression line in red.

lower wind speed ensures the bloom of dinoflagellates. Furthermore, the correlation coefficient between diatom and wind speed is positive in the Gulf of Finland and Gulf of Riga.

There are other physical factors in the sea that also have an effect. The results of the model show that due to spring circulation in the Gulf of Finland, more salty water of the surface layer (up to 10m) flows from the Baltic Proper into this Gulf. Salt tongue representation in the surface layer from 3 to 10m was especially clear in April 2013. The effect of the inflow is such that the correlation with wind speed in the Gulf of Finland becomes reversed in areas where fresher Baltic Proper water exists. In the Gulf of Riga, the inflow of fresh water and its circulation changed the direction of correlation. The southern part of the Baltic Proper has no correlation with the wind speed as it has had a strong upwelling during April in all the years studied and in the central part of Baltic Proper, phytoplankton values remain low for a long time.

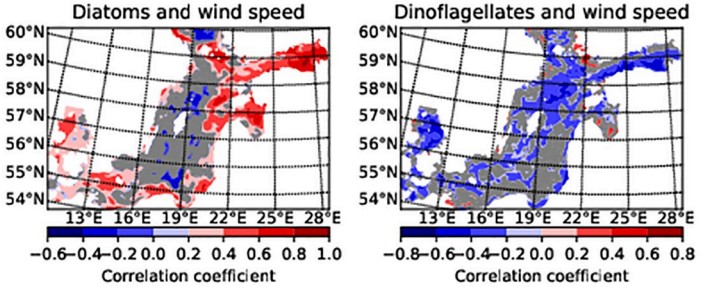

**Fig 12. The correlation coefficient (R) between a 5-day moving average wind speed and phytoplankton biomass.** The modelled dinoflagellates, diatoms, and ERA-5 wind on 5 April to 5 May 2010, 2011 and 2013 have been considered. N = 90 days, the critical value, Rcrit ($P < 0.05$) = 0.205, Rcrit ($P < 0.01$) = 0.267. The correlation in the red and blue areas is statistically significant ($P < 0.05$). Grey is statistically insignificant.

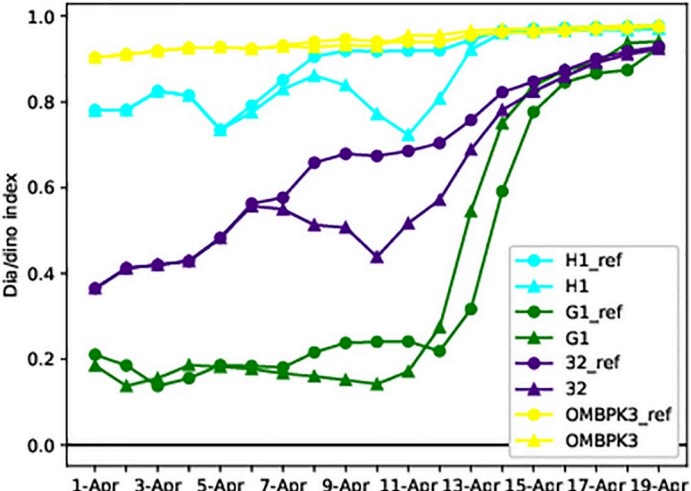

**Fig 13. Increase of dinoflagellates due to the wind reduction.** Dia/dino index in the reference run is represented by circles, and by triangles in the wind reduction scenario.

## Modelling of wind reduction scenario

We studied that low wind speed favours the accumulation of dinoflagellates which allows them to outcompete the faster growing diatoms. Using the wind reduction and reference scenarios we tested the influence of wind on modelled phytoplankton spring bloom. From the results obtained, it can be seen that the concentration of dinoflagellates in the euphotic layer increased due to wind reduction (Fig 13). G1 values were low in the beginning of April because the area was partly covered in ice while the stations H1, G1, OMBPK3 were ice-free. The environmental status of station 32 changed (dia/dino index threshold value is 0.5) in the euphotic layer during the wind reduction. However, the change occurs only in the surface layer of the sea, where the index value is greater than 0.5, and therefore the diatoms predominate. Station OMBPK3 is located in an area where circulation caused by dominant winds brings the bottom layers to the surface of the water (upwelling), thereby leaving the value of the index unchanged.

The Dissolved Inorganic Nitrogen (DIN) and Dissolved Inorganic Phosphorus (DIP) intakes at stations were analysed (Fig 14). In the case of a reference run, diatoms started developing and the dia/dino index was higher than the reduction run. Additionally, there were more diatoms (Fig 13) which consumed more nutrients. In the reduction scenario, when the physical conditions favour dinoflagellates, the nutrient intake is lower (Fig 14) and the DIN and DIP concentrations stay higher than during the reference run at all the stations.

## Discussion

Sea ice eliminates the effect of wind on water circulation, thus, affecting water mixing and currents; air does not come in contact with the water surface and sunlight is also reflected back into the atmosphere. We compared the results of numerical experiments performed to study the effect of the absence of ice on the water body in freezing temperatures and the presence of sea ice to assess the impact of sea ice. The results showed that the spring bloom of dinoflagellates appeared only in the sea area with thin ice (or low wind conditions) while diatoms dominated in ice-free water.

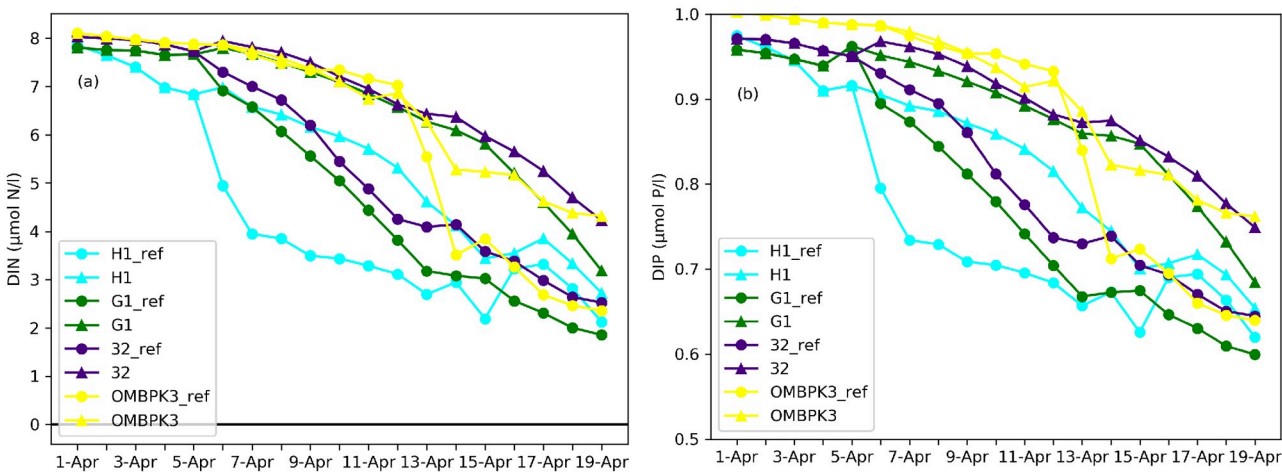

**Fig 14. Distribution of nutrients (DIN and DIP) in the reference and the wind reduction run.** The reference run is represented by circles, and by triangles in the wind reduction scenario.

Phytoplankton distribution is determined by the availability of nutrients, light, and temperature for growth, predation, sedimentation rate, and migration [43], as well as prevailing hydrodynamic conditions. The studied hydrodynamic conditions caused by wind speed, such as stratification, water mixing and currents, played a crucial role in the formation and location of phytoplankton biomass. In the modelling experiments, seasonal sea ice cover reduced wind-driven mixing and allowed the development of under-ice stratification (Figs 6 and 9). When water mixing is weak, large diatom cells sink while dinoflagellate cells remain in the surface layers because they exhibit neutral buoyancy. It may be noted that light can also be a limiting factor for dinoflagellates in the presence of diatoms, favouring the dominance of the latter.

In the present model nutrients most likely did not play a role because all simulations had the same nutrient input and initial conditions. However, the dynamics of DIN and DIP (Descriptor 5 of MSFD) were different for diatoms and dinoflagellates. When the predominant diatoms with a higher growth rate quickly consumed nutrients, it indicated the end of the spring bloom. When physical conditions for dinoflagellates were suitable, their nutrient intake was lower than diatoms and therefore the nutrients were available for longer in the euphotic zone. Diatoms lost their competitive advantage under sea ice and calm wind, as these conditions favoured sinking into the deeper layers of water where light was not available. Accordingly, the model results demonstrate how the changes in the sea ice or wind conditions drive changes in the Dia/Dino index.

The changes in dominance of these two phytoplankton classes affects the marine food web (Descriptor 4 of MSFD), and a low Dia/Dino index has been assumed by HELCOM to be an indicator of eutrophication (Descriptor 5 of MSFD) [2]. However, the results of the conducted modelling experiment show that during spring bloom low dia/dino index can be strongly influenced by physical factors and therefore should not be considered an indicator of eutrophication.

Potential reasons for dinoflagellate-dominated spring bloom becoming more frequent in the last decades are as follows:

1. Relatively thin ice: Studies have reported that ice conditions have gradually become milder over the years. However, the monitored observations for ice thickness around the Baltic Sea coast reveal both decreasing and increasing trends [13]. In the Gulf of Finland, a thinning

trend of -25 cm has been observed [44]. Sea ice in the middle of the Baltic Sea plays a crucial role in dinoflagellate-dominated spring bloom. A similar result was found by Klais et al. [20] which indicated that the local physical conditions (mild and variable winters) resulting from large-scale changes in weather patterns may be more crucial factors than nutrients in explaining the shift towards dinoflagellate dominance.

2.  Changeable weather in winter and spring: Since the 1990s, stratification collapse (strong mixing events) has frequently occurred in winters from October to April when saline and thermal stratification decreases [45]. Winters have become increasingly inconstant, with extremely high air temperatures for short periods [46]. High air temperatures cause sea ice to melt and reduce its thickness. When the temperature decreases again, the ice cover expands. The formation and closure of ice-free openings is a common phenomena in the Gulf of Finland [47–49].

Taking into account the results of this modelling experiment, the past, present, and the future of spring bloom in response to the reduction of ice cover due to climate change can be classified as follows:

1.  In the past, at the beginning of spring, the sea ice was so thick that the lack of light limited the growth of the phytoplankton [50]. Therefore, phytoplankton spring blooms occurred later, in the middle or end of May. Diatoms dominated in the southern part of the Baltic Sea where sea ice was absent.

2.  By the end of the 1980s, the light conditions in the central Baltic Sea improved due to reduced ice thickness, and both diatoms and dinoflagellates began to grow. Dinoflagellates gained a competitive advantage here because diatoms, being heavier, sank rapidly.

3.  According to future climate scenarios [14], the ice extent will be further reduced with shorter ice seasons and thinner ice cover. Completely ice-free winters are, however, unlikely to occur during the 21st century [51]. Most recent studies predict that dinoflagellates will dominate the spring bloom in the future. The modelling experiments conducted within this study, which only considered the changes in ice cover, demonstrated that loss of ice can be a factor promoting diatom dominance during spring bloom via changing the mixing regime.

Our modelling experiments were conducted over one growing season only. Longer-term model runs are necessary to elucidate whether the observed shifts towards diatom dominance during spring bloom are the initial reaction of the model to experimental perturbations, which will be negated by shifts in large-scale circulation and stratification caused by loss of sea ice, as well as alterations in nutrient availability and balance. Moreover, rather simplistic parameterisation of phytoplankton groups in the model applied is a limiting factor preventing more comprehensive answers regarding the future of phytoplankton spring blooms. To address this, in the future studies model formulation should be updated with the up-to-date understanding of the Baltic Sea phytoplankton ecology, especially formation of resting cysts, as well as consider wider diversity of phytoplankton functional groups.

## Conclusions

According to the study, ice conditions preceding the spring bloom influence the species composition of phytoplankton during spring bloom. In the presence of ice cover, which eliminates the effect of wind in the upper layer of water, the heavier particles (diatoms) sink below the euphotic zone (~ 10m) favouring proliferation of dinoflagellates. However, if diatoms remain

in the upper layer of water due to turbulent mixing induced by wind, they dominate the phytoplankton spring bloom.

The changing ice conditions are only one of the key factors directly affecting timing and composition of phytoplankton spring bloom within the Baltic Sea with cascading consequences for nutrient transfer and the entailing ecological dynamics.

The modelling experiments have demonstrated that the dia/dino index in spring is strongly influenced by physical factors, therefore further research is required to confirm the robustness of this index as an indicator of eutrophication [4, 8]. Since conclusions of this work are based on short-term model simulations, it should be regarded as an impetus for the future phytoplankton modelling studies that should test the long-term consequences of ice loss for phytoplankton bloom dynamics in the Baltic Sea using forecast scenario simulations.

## Supporting information

**S1 Fig. Diatom and flagellate concentrations (Chl mg/m$^3$) at station 14 in spring 2010.**
(a) Diatoms (no ice run); (b) diatoms (ice run); (c) dinoflagellates (no ice run); (d) dinoflagellates (ice run); (e) and (f) are water densities (kg/m$^3$) with ice run and no ice run. The water density is defined as the sigma density [(density(t,s,z)– 1000)] kg/m$^3$.
(TIFF)

**S2 Fig. Diatom and flagellate concentrations (Chl mg/m$^3$) at station 14 in spring 2011.**
(a) Diatoms (no ice run); (b) diatoms (ice run); (c) dinoflagellates (no ice run); (d) dinoflagellates (ice run); (e) and (f) are water densities (kg/m$^3$) with ice run and no ice run. The water density is defined as the sigma density [(density(t,s,z)– 1000)] kg/m$^3$.
(TIFF)

**S3 Fig. Diatom and flagellate concentrations (Chl mg/m$^3$) at station G1 in spring 2010.**
(a) Diatoms (no ice run); (b) diatoms (ice run); (c) dinoflagellates (no ice run); (d) dinoflagellates (ice run); (e) and (f) are water densities (kg/m$^3$) with ice run and no ice run. The water density is defined as the sigma density [(density(t,s,z)– 1000)] kg/m$^3$.
(TIFF)

**S4 Fig. Diatom and flagellate concentrations (Chl mg/m$^3$) at station G1 in spring 2011.**
(a) Diatoms (no ice run); (b) diatoms (ice run); (c) dinoflagellates (no ice run); (d) dinoflagellates (ice run); (e) and (f) are water densities (kg/m$^3$) with ice run and no ice run. The water density is defined as the sigma density [(density(t,s,z)– 1000)] kg/m$^3$.
(TIFF)

**S5 Fig. Diatom and flagellate concentrations (Chl mg/m$^3$) at station 32 in spring 2010.**
(a) Diatoms (no ice run); (b) diatoms (ice run); (c) dinoflagellates (no ice run); (d) dinoflagellates (ice run); (e) and (f) are water densities with ice run and no ice run. The water density is defined as the sigma density [(density(t,s,z)– 1000)] kg/m$^3$.
(TIFF)

**S6 Fig. Diatom and flagellate concentrations (Chl mg/m$^3$) at station 32 in spring 2011.**
(a) Diatoms (no ice run); (b) diatoms (ice run); (c) dinoflagellates (no ice run); (d) dinoflagellates (ice run); (e) and (f) are water densities with ice run and no ice run. The water density is defined as the sigma density [(density(t,s,z)– 1000)] kg/m$^3$.
(TIFF)

**S7 Fig. Diatom and flagellate concentrations (Chl mg/m$^3$) at station OMBPK3 in spring 2010.** (a) Diatoms (no ice run); (b) diatoms (ice run); (c) dinoflagellates (no ice run);

(d) dinoflagellates (ice run); (e) and (f) are water densities with ice run and no ice run. The water density is defined as the sigma density [(density(t,s,z)– 1000)] kg/m$^3$.
(TIFF)

**S8 Fig. Diatom and flagellate concentrations (Chl mg/m$^3$) at station OMBPK3 in spring 2011.** (a) Diatoms (no ice run); (b) diatoms (ice run); (c) dinoflagellates (no ice run); (d) dino-flagellates (ice run); (e) and (f) are water densities with ice run and no ice run. The water density is defined as the sigma density [(density(t,s,z)– 1000)] kg/m$^3$.
(TIFF)

**S1 Table. Peak of chlorophyll-a in April 2013.** Data from Baltic Nest, ICES and satellite data from Copernicus database.
(DOCX)

**S2 Table. The table shows dia/dino index and records stating the existence of ice.** Phytoplankton events based on observations. Cases, which developed under ice cover, as illustrated in figures, have been marked with an asterisk. Stations location A (58.35˚N;18.14˚E) and B (57.3˚N;20.1˚E). Ice concentration for up to 10 days before measurements.
(DOCX)

# Acknowledgments

I (OP) owe gratitude to the JRC Modeling Team for inspiring, teaching, and encouraging me. You have led me wisely and helped me transition from the world of physics to ecosystems.

The authors thank to the developers of GETM, GOTM and FABM as well. Thank you very much for your tips and for directing OP to the necessary measurement data, Norbert Wasmund, Susanne Feistel, Monika Quinones Winder, Inga Lips, Andres Jaanus, Natalja Kolesova. Thank you Irina Olenina for your cheerful and warm attitude and for your help in preparing the data.

# Author Contributions

**Conceptualization:** Ove Pärn, Gennadi Lessin, Adolf Stips.

**Data curation:** Ove Pärn.

**Formal analysis:** Ove Pärn.

**Funding acquisition:** Adolf Stips.

**Methodology:** Ove Pärn, Gennadi Lessin, Adolf Stips.

**Project administration:** Adolf Stips.

**Resources:** Adolf Stips.

**Supervision:** Gennadi Lessin, Adolf Stips.

**Validation:** Ove Pärn.

**Visualization:** Ove Pärn.

**Writing – original draft:** Ove Pärn.

**Writing – review & editing:** Gennadi Lessin, Adolf Stips.

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
