## [Decision Letter · Decision Letter 0]

26 Nov 2020

PONE-D-20-34799

Effects of sea ice and wind speed on Phytoplankton spring bloom in Central and Southern Baltic Sea

PLOS ONE

Dear Dr. Pärn,

Thank you for submitting your manuscript to PLOS ONE. After careful consideration, we feel that it has merit but does not fully meet PLOS ONE’s publication criteria as it currently stands. Therefore, we invite you to submit a revised version of the manuscript that addresses the points raised during the review process.

Both the reviewers have offered critical comments and have suggested major revision. The methodology followed and the inferences are also questioned. In view of such a review, though a major revision is suggested, authors need to convincingly answer the comments and take up much needed appropriate revision, for a fresh round of review.

We look forward to receiving your revised manuscript.

Kind regards,

Arga Chandrashekar Anil, Ph. D., D. Agr.,

Academic Editor

PLOS ONE

Journal Requirements:

2. We note that there are some minor issues with English grammar and language usage. Such examples include, but are not limited to: "In recent years have been mainly mild winter." (line 101). While we do not feel that these are prevalent enough to necessitate the use of a professional editing service, we suggest that you have a native English speaker review and edit the text for you.

3.Thank you for stating the following financial disclosure:

 [The funders had no role in study design, data collection and analysis, decision to publish, or preparation of the manuscript.].

4. We note that [Figure(s) 10 and 12] in your submission contain map images which may be copyrighted. All PLOS content is published under the Creative Commons Attribution License (CC BY 4.0), which means that the manuscript, images, and Supporting Information files will be freely available online, and any third party is permitted to access, download, copy, distribute, and use these materials in any way, even commercially, with proper attribution. For these reasons, we cannot publish previously copyrighted maps or satellite images created using proprietary data, such as Google software (Google Maps, Street View, and Earth). For more information, see our copyright guidelines: http://journals.plos.org/plosone/s/licenses-and-copyright.

1.    You may seek permission from the original copyright holder of Figure(s) [10 and 12] to publish the content specifically under the CC BY 4.0 license. 

Reviewers' comments:

Reviewer's Responses to Questions

**Comments to the Author**

1. Is the manuscript technically sound, and do the data support the conclusions?

Reviewer #1: Yes

Reviewer #2: Partly

2. Has the statistical analysis been performed appropriately and rigorously? 

Reviewer #1: Yes

Reviewer #2: I Don't Know

3. Have the authors made all data underlying the findings in their manuscript fully available?

Reviewer #1: Yes

Reviewer #2: Yes

4. Is the manuscript presented in an intelligible fashion and written in standard English?

Reviewer #1: Yes

Reviewer #2: Yes

5. Review Comments to the Author

Reviewer #1: The manuscript titled “Effects of sea ice and wind speed on Phytoplankton spring bloom in Central and Southern Baltic Sea” by Parn et al. describes a model study of the phytoplankton spring bloom composition in the Central and Southern Baltic Sea in relation to ice cover. Experiments with an established hydrodynamic–biogeochemical model (GETM + ERGOM) lead the authors to an interesting conclusion that in years with sea ice - dinoflagellates dominate in the phytoplankton biomass and, vice versa, in years with no sea ice - diatoms dominate. The authors then extrapolate that into the future when diatoms are expected to dominate as ice will almost certainly either disappear or become rare in the Southern and Central Baltic Sea due to climate change. These conclusions are certainly interesting and even intriguing. The explanations given by the authors make sense but it remains puzzling for the world ocean where many areas never experience ice but have with huge dinoflagellate blooms and, on the contrary, areas of the Arctic and Antarctic oceans with ice are often dominated by diatoms. It is difficult to understand why the Baltic Sea would be such a peculiar case. In my opinion the modeling study has been performed at high professional level and I cannot find any technical problems that would be producing those intriguing results. However, it is quite possible that the inherently crude parametrization of the dynamics of diatom and dinoflagellate populations leads to these model results. Natural diatom and dinoflagellate populations have huge variations in their characteristics, e.g. cell size, growth and sinking speed, vertical mobility, etc. and parameters fixed to some estimates may not lead to realistic outcomes of the model.

The manuscript is definitely too long and looks more like a report. The text should be shortened and the flow of text improved. It has a fair share of typos and omissions. The following is just a sample.

Line 26, Kili Bay is in Taiwan. Do you mean the Kiel Bay?

Fig 1 does not seem to have a caption.

Reviewer #2: Review of Pärn et al Effects of sea ice and wind speed on Phytoplankton spring bloom in Central and Southern Baltic Sea.

The paper is a model study that investigates the spring bloom in the Baltic Sea, and in particular focuses on the shifting dominance between diatoms and dinoflagellates. There has been a shift over the past decades from diatom dominance towards more dinoflagellates during this period. The conclusion drawn from the paper suggests diatoms dominate the spring bloom when there is thick ice cover, whereas dinoflagellates dominate when the ice cover is thin. They also argue that diatoms will dominate when the ice cover is completely removed something that might take place in a warmer future.

I am not a modeler and cannot go into the details behind the model, but I found the paper interesting and worthy of publication although I am not sure about the general conclusion and would like to see some revision in that respect.

Getting the model to work with the slower growing dinoflagellates can be difficult, and it seems they had to increase the growth rate above what is published under optimal conditions for dinoflagellates. Also, large parts of the central Baltic Sea is very often without ice and still dinoflagellates seem to dominate, see e.g. Fig 4 in Klais et al 2011. So the distribution of dinoflagellate dominance extends from typically ice covered to not ice covered.

What I think is something missing in the model is the fact that the spring bloom is driven by the inflow of freshwater creating the initial stratification rather than thermal stratification which requires longer time for the surface layer to warm up. I would suggest to play around with this in the model, or at the very least bring up the limitations in the model in the discussion as it clearly is not well able to capture the dinoflagellate dominance in the southern and central parts of the Baltic Sea that is typically ice free every winter.

Minor comments

L 75: Could add a line explaining that diatoms are non-motile and sink quicker whereas dinoflagellates are motile and can position themselves vertically.

L 79 Most of dinoflagellates sink in the form of cysts, otherwise vegitative cells tend to lyse before reaching the seafloor and still diatoms is probably the more quickly sinking group.

L 81: This is only in the case of the resting cysts that are not decomposed in the sediment

L84-97: The difference in growth rate is the same between dia and dinos in the Baltic Sea (Spilling and Markager 2008 - https://doi.org/10.1016/j.jmarsys.2006.10.012). What is likely a factor is timing of requirement (Kremp et al 2008) and stratification. What is different in the Baltic Sea from many other coastal regions is the massive freshwater outflow at spring and salinity stratification rather than thermal stratification playing an effect (Stipa 2002 & 2004) which is important for the start of the spring bloom.

L123. 0.7 d-1 is likely too high, 0.4 d-1 is more realistic

L124 ranging 0.6 to 1.4 d-1 depending on species

L313: In reality dinoflagellates are actually the dominating group at least in station 32.

L440-41: no they actually swim

L477-479: sounds like there was no spring bloom in the past

L483-486: There is ample evidence dinoflagellates dominate even under ice free winters so I do not buy this conclusion, a more interesting question is why the model gets it wrong.

Some new and relevant citations:

Hjerne, Olle, Susanna Hajdu, Ulf Larsson, Andrea Downing, and Monika Winder. "Climate driven changes in timing, composition and size of the Baltic Sea phytoplankton spring bloom." Frontiers in Marine Science 6 (2019): 482.

Lipsewers, Tobias, Riina Klais, Maria Teresa Camarena-Gómez, and Kristian Spilling. "Effects of different plankton communities and spring bloom phases on seston C: N: P: Si: chl a ratios in the Baltic Sea." Marine Ecology Progress Series 644 (2020): 15-31.

Camarena‐Gómez, María Teresa, Clara Ruiz‐González, Jonna Piiparinen, Tobias Lipsewers, Cristina Sobrino, Ramiro Logares, and Kristian Spilling. "Bacterioplankton dynamics driven by interannual and spatial variation in diatom and dinoflagellate spring bloom communities in the Baltic Sea." Limnology and Oceanography (2020).

6. PLOS authors have the option to publish the peer review history of their article (what does this mean?). If published, this will include your full peer review and any attached files.

Reviewer #1: No

Reviewer #2: No

---

## [Author Response · Author response to Decision Letter 0]

14 Jan 2021

Our sincere thanks for the relevant suggestions and comments that improve the quality of the manuscript and open up new perspectives.

 1.1 Reviewer #1: These conclusions are certainly interesting and even intriguing. The explanations given by the authors make sense but it remains puzzling for the world ocean where many areas never experience ice but have with huge dinoflagellate blooms and, on the contrary, areas of the Arctic and Antarctic oceans with ice are often dominated by diatoms. It is difficult to understand why the Baltic Sea would be such a peculiar case.

The focus of our manuscript is indeed on the influence of ice and wind. However, the mechanism of dominance of diatoms or dinoflagellates depends on many other factors such as water density (salinity, temperature), circulation characteristics (like upwelling), wind (water turbulent mixing), nutrient concentrations and balance, as well as phytoplankton physiology. 

Example, 3-4 days of wind silence (wind speed <5 m / s) can be very favorable for dinoflagellates.

We strongly agree that there are open points for discussion and further research (specifically multi-year and climate scenario simulations) and ecosystem model development are needed for parametrization of phytoplankton. 

1.2 Reviewer #1: “ However, it is quite possible that the inherently crude parametrization of the dynamics of diatom and dinoflagellate populations leads to these model results. Natural diatom and dinoflagellate populations have huge variations in their characteristics, e.g. cell size, growth and sinking speed, vertical mobility, etc. and parameters fixed to some estimates may not lead to realistic outcomes of the model.”

We fully agree with the reviewer that there are huge variations in the phytoplankton community. However, the model is inevitably a highly simplified representation of reality, where dominant types of phytoplankton are grouped according to similar characteristics (ref ERGOM papers). Updating the model according to up-to-date knowledge on phytoplankton ecology and physiology (including production of cysts), and possibly diversifying functional types of phytoplankton to include higher variability, are necessary to improve its performance in the future.

1.3 Reviewer #1: The manuscript is definitely too long and looks more like a report.

We would like to thank the reviewer for his comment and pointing this out. 

The paragraph “Ice Conditions during Spring Bloom” has now been removed from the paper to shorten its overall length.

We have reformulated the discussions and have specified the scope of the paper.

2.1 Reviewer #2: Getting the model to work with the slower growing dinoflagellates can be difficult, and it seems they had to increase the growth rate above what is published under optimal conditions for dinoflagellates.

L123. 0.7 d-1 is likely too high, 0.4 d-1 is more realistic

We agree with the reviewer that the values of the phytoplankton parameters need further improvement and the development of the model is in progress. 

When we used 0.4 d-1 in the model, the dia/dino index in the results obtained was not a close match with the results of Wasmund [1]. However, after changing the value to 0.7 d-1, we were able to obtain results which were a closer match to the results of Wasmund.

Therefore, we have used the value of the Ecological Regional Ocean Model (IOW ergom) 0.7 [2] which is used by the The Leibniz Institute for Baltic Sea Research, Warnemünde.

2.2 Reviewer #2: “Also, large parts of the central Baltic Sea is very often without ice and still dinoflagellates seem to dominate, see e.g. Fig 4 in Klais et al 2011. So the distribution of dinoflagellate dominance extends from typically ice covered to not ice covered.” 

Diatoms need less light and have a higher growth rate in the model, so they are more dominant. According to the model, dinoflagellates can only grow if the concentration of diatoms is low enough. This allows more light on the sea surface leading to the dominance of dinoflagellates. This is a situation that can exist in the central parts of the Baltic Sea. 

Another situation that can occur is when diatoms sink deep from the surface. Such conditions occur when there is ice in the area or when the daily average wind speed is less than 5 m/s. The circulation is also important.

In view of the foregoing, despite the central Baltic Sea often being without ice, dinoflagellates still seem to dominate because of the low wind speed that is not fully mixing the sea surface layers and ensuring that diatoms sink to the bottom. Also, the concentration of the diatoms is principally smaller (compared to the coastal areas) allowing the dinoflagellates to absorb light creating optimal conditions for their growth and dominance.

2.3 Reviewer #2: “L483-486: There is ample evidence dinoflagellates dominate even under ice free winters so I do not buy this conclusion, a more interesting question is why the model gets it wrong.”

The reviewer is correct that the algal spring bloom in the Baltic Sea was often characterised by the dominance of dinoflagellates over diatoms. However, in our experiments, in ice free open sea areas, often diatoms were found to dominate. On the other hand, the stratification caused in the water due to several variables would lead to a situation favouring the dominance of dinoflagellates, but this is potentially more important for coastal regions.

We understand that the model is not perfect but it has been reliable in the experiments we have conducted so far. However, this paper would give impetus to further the development of the model. 

2.4 Reviewer #2: “What I think is something missing in the model is the fact that the spring bloom is driven by the inflow of freshwater creating the initial stratification rather than thermal stratification which requires longer time for the surface layer to warm up.”

Yes, freshwater runoff has importance for the spring bloom. Inflow of freshwater is considered in the model and it plays a role with regard to nutrient input and stratification. However, in our modelling experiment we did not analyse the effects of freshwater inputs separately, focusing on integrated change in the upper water column dynamics due to lack of ice. As we were only simulating a single spring bloom event, the effect of freshwater inputs might have been not pronounced, and would be more apparent in long-term simulations, which we are currently considering as a follow-up to this work.

2.5 Reviewer #2: “L84-97: The difference in growth rate is the same between dia and dinos in the Baltic Sea (Spilling and Markager 2008 - https://doi.org/10.1016/j.jmarsys.2006.10.012). 

Many thanks for the relevant article references. 

In the model we have used different growth rates for diatom and dinoflagellate, 0.7 d-1 and 1.3 d-1 respectively. We have increased the growth rate of dinoflagellate from 0.4 to 0.7 [2, Neumann 2002]. According to the reference, it seems that a reduction in the growth rate of the diatom would have been more justified. 

2.6 Reviewer #2: What is likely a factor is timing of requirement (Kremp et al 2008) and stratification. What is different in the Baltic Sea from many other coastal regions is the massive freshwater outflow at spring and salinity stratification rather than thermal stratification playing an effect (Stipa 2002 & 2004) which is important for the start of the spring bloom.

Our model does take into account the outflow of freshwater. However, our modelling experiment with and without ice cover demonstrates that presence/absence of ice as an important factor shaping the composition of phytoplankton spring bloom. We certainly agree that there are other factors at play which will influence these dynamics in the future, including changes in atmospheric forcing, runoff and nutrient balance. We have changed relevant parts of the text to clarify this. 

2.7 Reviewer #2: “L 75: Could add a line explaining that diatoms are non-motile and sink quicker whereas dinoflagellates are motile and can position themselves vertically.”

Thank you, we changed the sentence in Introduction:

Wind is one of the factors governing vertical mixing within the euphotic layer ( ~9.6 m in Baltic, [29]) and in stagnant water, the diatoms, being non-motile and heavier, sink quickly, whereas the dinoflagellates are motile and can therefore position themselves vertically. 

2.8 Reviewer #2: L313: In reality dinoflagellates are actually the dominating group at least in station 32.

Phytoplankton composition has a great deal of variability, both in space and time. 

For example, on 14 and 15 March 2002, 14 measurements were made in a small sea area (data from Irina Olenina), 20⁰5’-21⁰01’; 55⁰39’-56⁰01’ (not too far from station 32). 

The mean biomass of the diatoms was 1024µG/L and the standard deviation was 880µG/L. The maximum value was 3425µG/L and the minimum was 71µG/L. The mean value of the dinoflagellates was 38µG/L. 

However, only a month later, the mean value of the dinoflagellates was 972µG/L, (std= 915µG/L, min=176, max = 2710), while the mean value of the diatoms was 420µG/L (std=519, min 24, max=1308). 

Therefore, it can be seen that one station cannot be fairly representative of which phytoplankton group is dominant during bloom when the measurements are taken only 1 or 2 times during spring.

2.9 Reviewer #2: L440-41: no they actually swim

Agreed; however, for simplicity, the model formulation does not take the ability of dinoflagellates to swim into account. We believe this assumption is justified given their slow speed of movement and the spatial resolution of the model applied in this study.

2.10 Reviewer #2: L477-479: sounds like there was no spring bloom in the past

We agree that the sentence was not clearly formulated. We have changed this sentence:

1. In the past, at the beginning of spring, the sea ice was so thick that the lack of light limited the growth of the phytoplankton [50]. Therefore, phytoplankton spring blooms occurred later, in the middle or end of May. Diatoms dominated in the southern part of the Baltic Sea where sea ice was absent. 

References

1. Wasmund N. The diatom/dinoflagellate index as an indicator of ecosystem changes in the baltic sea. 2. historical data for use in determination of good environmental status. Frontiers in Marine Science. 2017 Jun 8;4:153.

2. Neumann T, Fennel W, Kremp C. Experimental simulations with an ecosystem model of the Baltic Sea: a nutrient load reduction experiment. Global biogeochemical cycles. 2002 Sep;16(3):7-1.

---

## [Decision Letter · Decision Letter 1]

1 Feb 2021

Effects of sea ice and wind speed on Phytoplankton spring bloom in Central and Southern Baltic Sea

PONE-D-20-34799R1

Dear Dr. Pärn,

We’re pleased to inform you that your manuscript has been judged scientifically suitable for publication and will be formally accepted for publication once it meets all outstanding technical requirements.

Kind regards,

Arga Chandrashekar Anil, Ph. D., D. Agr.,

Academic Editor

PLOS ONE

Additional Editor Comments (optional):

Reviewers' comments:

Reviewer's Responses to Questions

**Comments to the Author**

1. If the authors have adequately addressed your comments raised in a previous round of review and you feel that this manuscript is now acceptable for publication, you may indicate that here to bypass the “Comments to the Author” section, enter your conflict of interest statement in the “Confidential to Editor” section, and submit your "Accept" recommendation.

Reviewer #2: All comments have been addressed

2. Is the manuscript technically sound, and do the data support the conclusions?

Reviewer #2: Yes

3. Has the statistical analysis been performed appropriately and rigorously? 

Reviewer #2: N/A

4. Have the authors made all data underlying the findings in their manuscript fully available?

Reviewer #2: Yes

5. Is the manuscript presented in an intelligible fashion and written in standard English?

Reviewer #2: Yes

6. Review Comments to the Author

Reviewer #2: I am happy with the response to the review and the changes done. There are still a few typos so please read carefully through it before the final version goes to print.

7. PLOS authors have the option to publish the peer review history of their article (what does this mean?). If published, this will include your full peer review and any attached files.

Reviewer #2: No

---

## [Editor Report · Acceptance letter]

4 Feb 2021

PONE-D-20-34799R1 

Effects of sea ice and wind speed on Phytoplankton spring bloom in Central and Southern Baltic Sea 

Dear Dr. Pärn:

I'm pleased to inform you that your manuscript has been deemed suitable for publication in PLOS ONE. Congratulations! Your manuscript is now with our production department. 

Kind regards, 

on behalf of

Professor Arga Chandrashekar Anil 

Academic Editor

PLOS ONE